# Predictability: A new distinguishing feature of cancer?

Ofer N. Gofrit[1]*, Ariel Aviv[2]

1 Department of Urology, Hadassah Hebrew University Hospital, Jerusalem, Israel, 2 Department of Hematology, Ha'Emek Medical Center, Afula, Israel

* ogofrit@gmail.com

**Data Availability Statement:** All data is available in supp. material 1

**Funding:** The author(s) received no specific funding for this work.

## Abstract

Cancer is a consequence of stochastic (mutations, genetic, and epigenetic instabilities) and deterministic (evolutionary bottlenecks) events. Stochastic events are less amenable to prediction, whereas deterministic events yield more predictable results. The relative contribution of these opposing forces determines cancer predictability, which affects the accuracy of our prognostic predictions and is critical for treatment planning. In this study, we attempted to quantify predictability. The predictability index (PI) was defined as the median overall-survival at any time point divided by the standard error at that time. Using data obtained from the SEER program, we found striking differences in the PI of different tumors. Highly predictable tumors were malignancies of the breast, thyroid, prostate, and testis (5-year PI of 3516, 1920, 1919, and 1805, respectively). Less predictable tumors were colorectal, melanoma, and bladder (5-year PI of 1264, 1197, and 760, respectively). Least predictable were pancreatic cancer and chronic myelogenous leukemia (5-year PI of 129, and 42). PI decreased during follow-up in all examined tumors and showed sex differences in some cases. Thyroid cancer was significantly more predictable in women (5-year PI of 2579 vs. 748, p = 0.00017) and bladder cancer more predictable in men (5-year PI of 723 vs. 385, p = 0.012), Predictability is a potentially new distinguishing feature of malignancy. This study sheds light on prognostic accuracy and provides insight into the relative roles of stochastic and deterministic forces during carcinogenesis.

## Introduction

Somatic mutation theory, the prevailing paradigm of carcinogenesis, suggests that cancer results from the selection of cells genetically mutated at critical sites, that is, oncogenes and tumor suppressor genes [1, 2]. Development of cancer is a two-phase process: initiation by extrinsic or intrinsic carcinogen action upon DNA, a completely stochastic process (no two tumors are identical in genotype), and promotion: cells' gain of proliferative, invasive, and metastatic capabilities, a process that includes both stochastic and deterministic elements.

Mutations are independent and stochastic. As cancer progresses, genetic instability, structural chromosomal aberrations, and epigenetic changes augment the chaotic elements of carcinogenesis. However, there are multiple deterministic steps during carcinogenesis, the

**Competing interests:** The authors declare no conflicts of interest.

bottlenecks that the tumor must cross to thrive, including angiogenic shift, evasion of apoptosis, self-sufficiency in growth signals, tissue invasion, etc [3]. Each bottleneck deterministically selects clones with a higher proliferative capacity and better ability to thrive by escaping normal control mechanisms. Therapeutic efforts further add evolutionary pressure towards the selection of resistant clones, adding determinism to cancer progression.

Other theories of carcinogenesis have proposed an even more significant role for the deterministic processes. For example, the atavistic theory suggests that carcinogens induce stochastic DNA damage, which in turn activates a well-orchestrated rescue toolkit, which is reminiscent of our Metazoa 1.0 unicellular eukaryotic ancestor. The vital functions of these primordial organisms, namely, survival and proliferation, are the core features of cancer. According to this theory, cancer is a single or a series of atavistic events originating from our phylogenetic history, and thus should behave in a highly deterministic manner [4].

Therefore, theories of carcinogenesis suggest the involvement of both stochastic and deterministic events. Because deterministic forces are expected to result in predictable consequences, and stochastic forces are unpredictable, we hypothesized that the relative magnitude of these forces can be estimated from the predictability of cancer progression if only we can measure it. We hypothesized that predictability can be measured and can provide vital information to the patient and insights to the events occurring during cancer initiation and progression. Since the accuracy of patients' prognosis is expressed by the dispersion of values around the median overall survival (lower and upper confidence intervals) we used this information for measuring cancer predictability.

## Materials and methods

Predictability index (PI) was defined as follows: patients' median overall survival (OS) at any time point divided by the dispersion of values around the median- the standard error (SE). PI = median OS/SE. PI was calculated from data obtained from the Surveillance, Epidemiology, and End Results (SEER) program [5]. This database covers approximately 28% of the population of the United States. By 2023, 1,958,310 new cancer cases were included in the SEER program. Because the database is open to the public and does not require patient-informed permission, institutional review approval was not necessary for our study. The SEER program provides overall survival (OS) and 95% lower and upper confidence intervals (CIs) over 10 years of follow-up for all common cancer types, and a breakdown according to sex. Standard error (SE) was calculated as follows: SE = (upper 95%CI-lower 95%CI)/3.92. Clearly, PI is not a constant value and it changes over time after diagnosis. PIs between men and women were compared using two-tailed t-tests for paired samples and a p-value < 0.05 was considered significant. PI should reflect the balance between stochastic and deterministic events during carcinogenesis.

## Results

The complete dataset is available in the S1 Data. PI at 5-years after diagnosis (PI5) of various cancer types is presented in Table 1. Malignancies of the breast, thyroid, prostate, testis, colorectum, and melanomas were found to have a highly predictable course (PI5 > 1000). Tumors of the bladder, lymphomas, cervix, uterus, lungs, sarcomas, and ovaries showed intermediate predictability (PI5 between 300 and 1000). Multiple myeloma, leukemia, and pancreatic cancer showed low predictability (PI5 lower than 300). Chronic myelomonocytic leukemia and acute monocytic leukemia-M5 showed extremely low predictability (PI5s of 42 and 36, respectively).

PI decreased over time for all tumor types (Fig 1). For example, the PI of prostate cancer decreased from 3888 one year after diagnosis to 1905 at 10 years, bladder cancer PI decreased

**Table 1. Five-year predictability indexes (PI5) according to cancer type.** The types are ordered in decreasing PIs.

| Cancer Type | New cases (2023) | 5-year overall survival (%) | S.E. | 5-year Predictability Index |
|---|---|---|---|---|
| All types | 1,958,310 | 66.8 | 0.025 | 2619 |
| Breast | 297,790 | 89.7 | 0.025 | 3516 |
| Thyroid | 153,020 | 98.0 | 0.051 | 1920 |
| Prostate | 288,300 | 97.9 | 0.051 | 1919 |
| Testis | 9190 | 92.1 | 0.051 | 1805 |
| Colo-rectum | 153,020 | 64.5 | 0.051 | 1264 |
| Melanoma | 97,610 | 91.6 | 0.102 | 1197 |
| Bladder | 82,290 | 77.5 | 0.102 | 760 |
| Non-Hodgkin lymphoma | 80,550 | 70.8 | 0.102 | 694 |
| Hodgkin's disease | 8,830 | 86.0 | 0.153 | 561 |
| Cervix | 13,960 | 68.0 | 0.153 | 444 |
| Chronic myeloproliferative disorder | | 84.7 | 0.204 | 415 |
| Uterus | 66,220 | 80.3 | 0.204 | 394 |
| Lung | 238,340 | 19.9 | 0.051 | 390 |
| Ovary | 19,710 | 47.8 | 0.128 | 374 |
| Stomach | 13,960 | 31.3 | 0.127 | 245 |
| Anus | 9,760 | 68.0 | 0.280 | 242 |
| Sarcoma | 13,400 | 65.7 | 0.204 | 321 |
| Multiple Myeloma | 35,730 | 51.4 | 0.178 | 289 |
| Acute lymphocytic leukemia | 6,540 | 68.2 | 0.255 | 267 |
| Chronic lymphocytic leukemia | 18,740 | 84.3 | 0.178 | 217 |
| Acute myeloid leukemia | 20,380 | 66.5 | 0.306 | 217 |
| Myelodysplastic Syndrome | | 40.9 | 0.204 | 200 |
| Bone & Joints | 3,970 | 98.2 | 0.357 | 190 |
| Chronic myeloid leukemia | 8,930 | 27.1 | 0.178 | 151 |
| Kaposi's sarcoma | | 70.3 | 0.484 | 145 |
| Pancreas | 67,050 | 9.9 | 0.076 | 129 |
| Chronic myelomonocytic leukemia | 8,930 | 25.8 | 0.791 | 42 |
| Acute monocytic leukemia-M5 | | 24.1 | 0.663 | 36 |

from 1748 to 551, colorectal cancer PI decreased from 1633 to 752, and melanoma PI decreased from 3806 to 705. The tumor with the highest drop in PI was chronic lymphocytic leukemia (4.8 folds), and the tumor with the smallest drop was breast cancer (1.16 folds). None of the tumors demonstrated an increase in PI, emphasizing that all tumors became less predictable over time.

PI showed sex differences in some tumors (Table 2 and Fig 2). Women showed significantly better predictability of thyroid cancer (PI5:2579 vs. 748, p = 0.00017) and melanoma (PI5:1015 vs. 903, p = 0.00017). Men showed better predictability of bladder cancer (PI5 of 723 vs. 385, p = 0.012), stomach cancer (PI5 of 184 vs. 146, $p = 1.4 \times 10^{-5}$), Kaposi's sarcoma (PI5 of 149 vs. 39, $p = 1.4 \times 10^{-8}$), and most leukemia types.

## Discussion

Predicting the clinical course of patients is extremely important in oncology. It provides crucial information to the patient and is critical for treatment planning and for comparison of trial results [6]. However, no attempt has been made thus far to quantify cancer predictability. We defined the PI and examined it using SEER data (Table 1 and S1 Data). A two-order

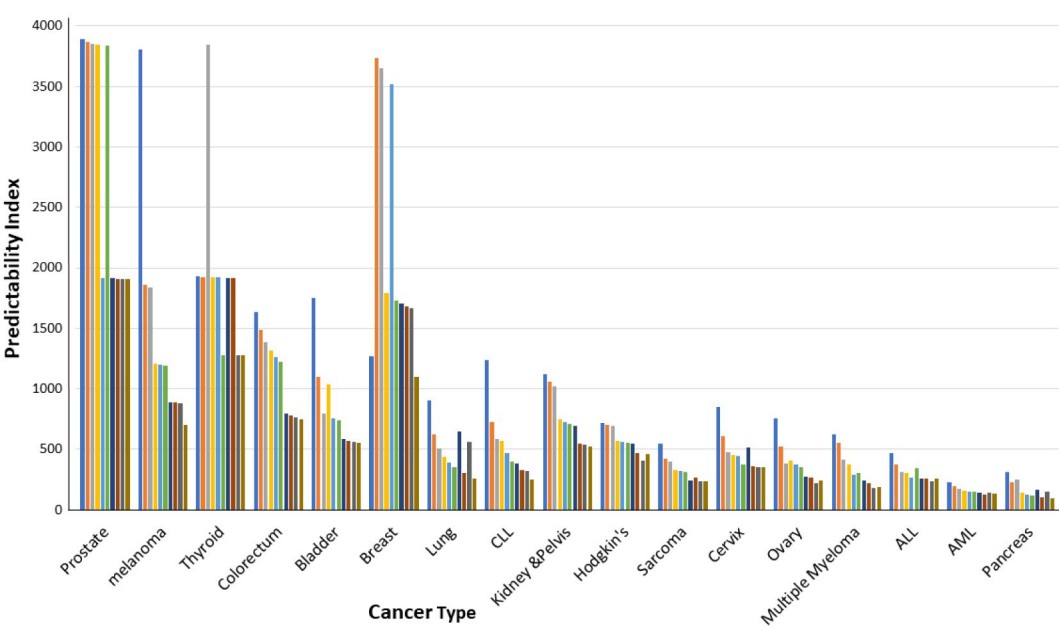

**Fig 1. Changes in predictability index in different tumor types according to time after diagnosis.** The predictability index of each tumor is presented from one year to ten years after diagnosis.

**Table 2. Comparisons of mean predictive indexes of women and men.** The predictive indexes over 10 years of follow-up were used to calculate the means presented. Cancer types are ordered according to increasing levels of statistical differences.

| Cancer Type | Mean Predictive Index Women | Mean Predictive Index Men | p Value |
|---|---|---|---|
| All types | 2481 | 2289 | 0.248 |
| Hodgkin's disease | 385 | 381 | 0.844 |
| Acute Myelogenous Leukemia | 115 | 115 | 0.790 |
| Non- Hodgkin's Lymphoma | 580 | 590 | 0.661 |
| Lung | 353 | 327 | 0.552 |
| Pancreas | 114 | 116 | 0.517 |
| Chronic myeloproliferative disorder | 334 | 279 | 0.462 |
| Myeloma | 234 | 203 | 0.439 |
| Myelodysplastic syndrome | 152 | 146 | 0.194 |
| Colorectum | 735 | 817 | 0.186 |
| Kidney & Pelvis | 538 | 620 | 0.089 |
| Melanoma | 1015 | 903 | 0.012 |
| Sarcoma | 225 | 252 | 0.027 |
| Chronic myelomonocytic Leukemia | 31 | 37 | 0.022 |
| Chronic Myelogenous Leukemia | 166 | 177 | 0.014 |
| Chronic lymphocytic Leukemia | 353 | 392 | 0.008 |
| Acute lymphocytic Leukemia | 196 | 227 | 0.004 |
| Thyroid | 2579 | 748 | 0.00017 |
| Bladder | 385 | 723 | 0.00016 |
| Acute monocytic Leukemia-M5 | 27 | 26 | 0.0001 |
| Stomach | 146 | 184 | $1.44 \times 10^{-5}$ |
| Kaposi's sarcoma | 39 | 149 | $5.9 \times 10^{-8}$ |

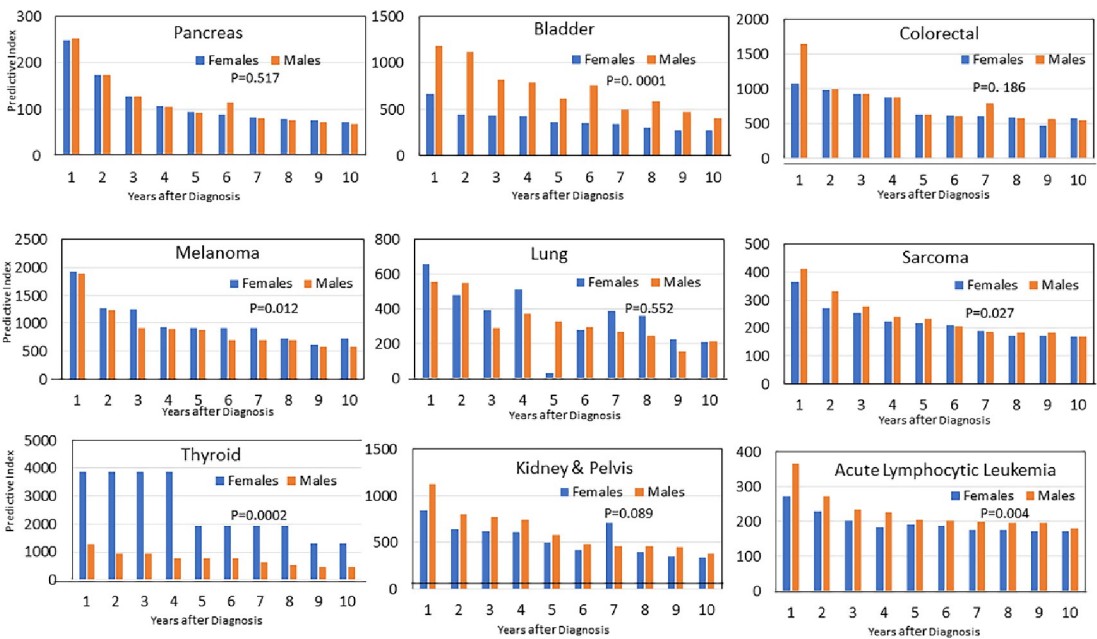

**Fig 2. Predictability index in men and women in selected tumors.**

magnitude difference in PI was found between different tumors. The most predictable tumor was breast cancer (PI5 of 2619) and the least predictable was acute monocytic leukemia-M5 (PI5 of 36). Other findings included a consistent drop in PI during the follow-up of all tumors, and sex differences in several tumors. How can these findings be explained? Several possible factors determine tumor predictability.

1. Heterogeneity of the tumor: More heterogeneous tumors are expected to behave less predictively.

2. The number of bottlenecks a tumor must pass during carcinogenesis. More bottlenecks imply higher predictability.

   Tumor heterogeneity includes four elements [7]:

a. Intratumoral heterogeneity: Every cell division introduces a few mutations, and more alterations are expected when the genes associated with genomic stability are mutated.

b. Intermetastatic heterogeneity: This results from different waves of metastases originating from different tumor clones and modes of metastatic spread, either linear or parallel [8].

c. Intrametastatic heterogeneity: Each metastasis is presumed to result from a single cell or small cluster of cells with similar founder mutations. However, as metastatic cells divide, they gain heterogeneity, similar to that in primary tumors.

d. Interpatient heterogeneity is due to individual germline and somatic mutations, and differences in the distribution and elimination of medications.

Tumor heterogeneity is related to the frequency of mutations, span of mutation frequencies, and order of mutation occurrence. Tumor population should be nearly homogeneous and predictable when mutation rate is low. When the rate is high, several parallel clones may develop simultaneously, decreasing clinical predictability [9]. Lawrence et al. published a list of somatic

mutation frequencies in multiple tumors [10]. This list bears some similarities to the predictability list in Table 1, but there are notable exceptions. Melanoma has the highest median mutation rate (100/Mb) but was in the top third of the predictability list (PI5 of 1197). AML has a low mutation rate (0.37/MB) and low predictability (5-year PI5 of 217). Lawrence et al. noted that the mutation rate did not completely explain tumor heterogeneity. For example, despite its relatively low mutation frequency, AML has a span of mutation frequencies of three orders of magnitude among different patients (0.01 to 10 Mb), which may explain the difficulty of predicting its course in different patients. In contrast, breast cancer, the most predictable tumor in the current study, was in the middle of the list regarding mutation frequency (median of 1/Mb); however, the span of mutation frequencies was narrow in this tumor, explaining its high predictability. This explanation does not hold for pancreatic cancer, which is also in the middle of the mutation frequency list (median of 1/Mb), has a narrow span of mutation frequencies but a very low PI. A different explanation will be proposed later for the low PI of this tumor.

The order of mutation occurrence is another source of diversity among patients. The best-studied case is myeloproliferative disorder in J*AK2 V617F* and *TET2* double-mutant patients. Patients with the J*AK2 V617F* first mutation are 10 years younger at diagnosis and have a higher probability of polycythemia vera and thrombotic events than patients with *the TET2* first mutation [11]. This mechanism likely involves different genetic and epigenetic changes induced by different initial mutations. In myeloproliferative disorders, both J*AK2 V617F* and *TET2* are known to alter the chromatin arrangement. Additionally, the progeny of the first mutation may create a microenvironment that drives different signaling responses to the second mutation. Mutation order is more difficult to study in solid tumors, but there is no reason to assume that the same cellular mechanisms do not apply there, increasing their diversity and decreasing their predictability.

Epigenetic changes can also increase cellular variability and thus decrease cancer predictability. These changes enable the development of genetically indistinguishable sub-populations with different sensitivity to systemic treatments. They can take the form of epigenetic heterogeneity (variability across cell population) that act as substrate for Darwinian selection or epigenetic plasticity (alterations in epigenetic state in response to external or internal stimuli with adaptive transcriptional, posttranslational, or metabolic plasticity) that act as a substrate for Lamarkian adaptation [12, 13]. Experimental models demonstrated that both transcriptional adaptation and posttranslational modifications can be found in acute myeloid leukemia, potentially increasing the variability of this disease and contributing to its low PI [14, 15].

As presented in the classical manuscript by Hanahan and Weiberg, all developing tumors must acquire several capabilities including self-sufficiency of growth signals, insensitivity to anti-growth signals, ability to evade apoptosis, limitless replicative potential, sustained angiogenesis, tissue invasion, and metastasis [3]. Obtaining each of these capabilities requires extremely complex machinery, which can be considered a variability-reducing bottleneck. Unlike solid tumors, liquid tumors such as leukemia have fewer bottlenecks to cross. They are not obliged to promote angiogenesis, they have a natural propensity for entering and exiting the bloodstream, and distant metastases formation is not a pre-requisite for patients' shortened survival. This may also explain the low predictability of sarcomas (PI5 = 321). These tumors originate in the stroma and have fewer bottlenecks to cross compared to carcinomas.

During follow-up, all the examined tumors showed decreasing predictability (Figs 1 and 2). This phenomenon can be attributed to an increase in mutation rate and in genomic instability as the cancer cell population increases [6]. Many cell divisions together with genomic instability are ideal for clonal diversity [16]. The inhomogeneity of metastatic microenvironments can also decrease the predictability [17]. This may also explain the low predictability of pancreatic

cancer (PI5 = 129). This tumor is commonly diagnosed late in its natural course, often with clinical metastases at initial diagnosis.

Another plausible explanation for the decrease in PI over time is that the longer patients survive, the more prone they are to adverse medical events not necessarily related to cancer, thereby increasing their chances of death from unpredictable causes. For example, deaths attributed to Hodgkin disease increase steadily in the first ten years after diagnosis and then plateau, while deaths from other causes relentlessly rise throughout life [18]. Additionally, patients with the shorter follow-up were diagnosed and treated more recently. They were subjected to more sensitive diagnostic tools, to more effective and less toxic therapies, and to better management of their comorbidities. Therefore, stage migration and improved care could have decreased prognosis variability and improv the PI of more recently diagnosed patients.

Sex disparities in PI were found in several tumor types (Table 2 and Fig 2). The most prominent differences were observed in bladder and thyroid malignancies. Thyroid cancer was significantly more predictable in women (PI5:1934 in women and 749 in men, p = 0.00017). Significant sex disparities in incidence and prognosis characterize this disease: it is three times more common in women; however, men have more aggressive disease at presentation, lower disease-free survival, and a higher mortality rate [19]. To date, no good explanation has been provided for this clinical disparity. One possible explanation for this phenomenon could be a detection bias. As women have more benign thyroid nodules, they are more likely to undergo a diagnostic work-up of the thyroid and be diagnosed with early thyroid cancer. Whereas men present later with more advanced disease explaining their lower PI. This is only a partial explanation because sex disparity is not limited to the first few years after diagnosis and lingers for at least 10 years after diagnosis (Fig 2). Bladder cancer showed the reverse trend. It was significantly more predictable in men (PI5 of 360 in women and 619 in men, p = 0.00016). Notably, clinical sex disparities are well-appreciated in this type of cancer. Men are affected three times more often, but women tend to present at a later stage and show higher mortality rates, even after adjusting for tumor stage and grade. Explanations for these differences are far from complete and include differences in carcinogen metabolism in the liver, immune responses, and the pivotal role of estrogenic receptors. In contrast to breast cancer, ERα is tumor-suppressive in the bladder, and ERβ promotes cancer initiation. Both receptors promote cancer progression. Similar to thyroid cancer in men, delayed diagnosis of bladder cancer often occurs in women [20–22]. Comparisons of somatic mutational data showed similar total mutation counts (median of 92 in females and 91 in males, p = 0.62), but significantly higher mutation rates in *ARID1A* and *NCOR1* in males and a non-significantly higher risk of mutations in *TP53* in females [23].

The PI model presented herein has several limitations. PI was based on the OS of patients. However, over a long follow-up period, other causes of death may have occurred, especially in elderly patients. A possible future analysis using progression free survival rather than OS might better reflect cancer predictability. Additionally, the SEER database does not differentiate between different cancer stages and includes patients with both localized and systemic malignancies in the same category. A parameter that considers cancer stages, comorbidity (which can decrease predictability, especially in elderly patients), medications, etc. should provide a more accurate measure of predictability.

## Conclusions

Predictability is a new and potentially defining feature of the malignant process. It can be quantified using a predictive index (PI). We found that different tumors showed remarkably different PIs. PI decreases during follow-up in all cancer types, and some tumors showed sex

disparity. PI is determined during carcinogenesis by the variability of tumor cells, host responses, and evolutionary bottlenecks that the tumor must undergo during its development. Studies in PI could potentially shed light on the accuracy of our prognostic ability and can provide insights into the relative magnitudes of stochastic and deterministic forces during pathogenesis.

## Supporting information

**S1 Data.**
(XLSX)

## Author Contributions

**Conceptualization:** Ofer N. Gofrit, Ariel Aviv.

**Data curation:** Ofer N. Gofrit.

**Formal analysis:** Ofer N. Gofrit, Ariel Aviv.

**Methodology:** Ofer N. Gofrit, Ariel Aviv.

**Writing – original draft:** Ofer N. Gofrit, Ariel Aviv.

**Writing – review & editing:** Ofer N. Gofrit, Ariel Aviv.

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
