## [Decision Letter · Decision Letter 0]

15 May 2024

PONE-D-24-05545Predictability: A New Distinguishing Feature of Cancer?PLOS ONE

Dear Dr. Gofrit,

Thank you for submitting your manuscript to PLOS ONE. After careful consideration, we feel that it has merit but does not fully meet PLOS ONE’s publication criteria as it currently stands. Therefore, we invite you to submit a revised version of the manuscript that addresses the points raised during the review process.

We look forward to receiving your revised manuscript.

Kind regards,

Sina Azadnajafabad, MD, MPH

Academic Editor

PLOS ONE

Journal Requirements:

"None"

Reviewers' comments:

Reviewer's Responses to Questions

**Comments to the Author**

1. Is the manuscript technically sound, and do the data support the conclusions?

Reviewer #1: Yes

Reviewer #2: No

2. Has the statistical analysis been performed appropriately and rigorously? 

Reviewer #1: Yes

Reviewer #2: No

3. Have the authors made all data underlying the findings in their manuscript fully available?

Reviewer #1: Yes

Reviewer #2: Yes

4. Is the manuscript presented in an intelligible fashion and written in standard English?

Reviewer #1: Yes

Reviewer #2: Yes

5. Review Comments to the Author

**Reviewer #1:** Thank you for the opportunity to review this interesting manuscript, which explores the variation in survival for different types of cancer. The study is original, and I am not aware of any similar undertaking. This line of research can highlight interesting clinical phenomena, and to inform both current clinical practice and future directions of research. I congratulate the authors on the idea to explore this interesting and provoking idea.

Major comment

- While I appreciate the idea, I think the methods can be improved. Genetic heterogeneity is not the only cause of variability in clinical outcomes. The SEER dataset includes demographics, tumor subtypes and stage data- all directly associated with outcome variability. Why not include these data in the analysis? Perhaps some of the variability might be explained by heterogeneity in cancer subtypes/stage at diagnosis? This variation in outcomes will not necessarily represent variation in predictability.

Minor comments

- The manuscript can benefit from scientific editing by a native English speaker, for grammar and clarity.

- I’m not sure why the authors chose the subjective term “predictability” and not a more neutral “variation” in describing their results.

- The authors should discuss the non-genetic factors which might influence patient outcome, which might be more variable in patients with some cancer types; various comorbidities, access to treatment issues etc. Genetics is a major factor, but not the only one.

**Reviewer #2:** The present article tries to prove that the survival of patients across different time points can be predicted and set a new carcinogenesis parameter

The drawbacks of such an assumption among other are

1) Stage migration due to more sophisticated tachniques during last decades

2) Different treatment efficacy during last decades

3) Different treatment toxicity during last decades

4) Different expected population survival due to the improvements in other specialties (e.g. cardiology) during last decades

6. PLOS authors have the option to publish the peer review history of their article (what does this mean?). If published, this will include your full peer review and any attached files.

Reviewer #1: No

Reviewer #2: No

---

## [Author Response · Author response to Decision Letter 0]

16 May 2024

Emily Chenette May 16, 2024 Editor-in-Chief PLoS One 

Dear Editor 

Thank you for considering the manuscript and for the reviewer's comment. Please find the enclosed response letter and the revised version.

We hope that you will find it interesting and suitable for publication in PLoS One.

Thanks

Ofer Gofrit 

Ariel Aviv

Reviewers' comments and responses:

Reviewer's Responses to Questions

Comment 1: Is the manuscript technically sound, and do the data support the conclusions?

Reviewer #1: Yes

Reviewer #2: No

Response: All the raw data (obtained from the SEER program) is available in Supp 1.xlsx. Calculation of the predictability index (PI) is as follows: PI=median overall survival/standard error. Standard error= (upper 95%CI-lower 95%CI)/3.92 (page 4, materials and methods). 

Comment 2: Has the statistical analysis been performed appropriately and rigorously?

Reviewer #1: Yes

Reviewer #2: No

Response: The study employs very simple statistics (only t-test comparing PI values of men and women).

Comment 3: Have the authors made all data underlying the findings in their manuscript fully available?

Reviewer #1: Yes

Reviewer #2: Yes

Response: None

Comment 4: Is the manuscript presented in an intelligible fashion and written in standard English?

Reviewer #1: Yes

Reviewer #2: Yes

Response: None

Reviewer #1: Thank you for the opportunity to review this interesting manuscript, which explores the variation in survival for different types of cancer. The study is original, and I am not aware of any similar undertaking. This line of research can highlight interesting clinical phenomena, and to inform both current clinical practice and future directions of research. I congratulate the authors on the idea to explore this interesting and provoking idea.

Major comment: While I appreciate the idea, I think the methods can be improved. Genetic heterogeneity is not the only cause of variability in clinical outcomes. The SEER dataset includes demographics, tumor subtypes and stage data- all directly associated with outcome variability. Why not include these data in the analysis? Perhaps some of the variability might be explained by heterogeneity in cancer subtypes/stage at diagnosis? This variation in outcomes will not necessarily represent variation in predictability.

Response: We wish to thank the reviewer for his kind words. The current manuscript is an initial attempt to turn the lights to predictability as a distinct feature of cancer. The proposed parameter "predictability index" is a preliminary attempt to quantify predictability. Adding more parameters, as the reviewer suggests, would certainly make it more accurate. We added these points to the discussion, however, the SEER program is not that accurate. It does not differentiate between cancer stages and does not consider comorbidity. We hope that future scientists will find interest in measuring cancer predictability and will come up with more accurate measures based on a more complete database as the reviewer suggests.

Minor comments

Comment 1: The manuscript can benefit from scientific editing by a native English speaker, for grammar and clarity.

Response: Any manuscript can be made better; but the current manuscript was revised by several native English speakers and by a professional scientific linguistic editor. Of course, any specific linguistic mistake can and should be corrected.

Comment 2: I’m not sure why the authors chose the subjective term “predictability” and not a more neutral “variation” in describing their results.

Response: Choosing an appropriate name was not an easy task. We wanted a term that will (quantitatively) reflect our ability to predict prognosis, therefore, we chose the term “predictability index”. 

Comment 3: The authors should discuss the non-genetic factors which might influence patient outcome, which might be more variable in patients with some cancer types; various comorbidities, access to treatment issues etc. Genetics is a major factor, but not the only one.

Response: The important contribution of non-genetic factors to tumor's heterogeneity was added to the manuscript. These factors, especially epigenetic changes enable the development of genetically indistinguishable sub-populations with different sensitivity to systemic treatments. These changes can take the form of epigenetic heterogeneity (variability across cell population) that act as substrate for Darwinian selection or epigenetic plasticity (alterations in epigenetic state in response to external or internal stimuli with adaptive transcriptional, posttranslational, and metabolic plasticity) that act as a substrate for Lamarkian adaptation (Bell 2020, Inde 2018). Experimental models demonstrated that both transcriptional adaptation and posttranslational modifications can be found in acute myeloid leukemia, potentially increasing the variability of this disease and contributing to its low PI (Bell 2019, Irish 2004). These points were added to the "Discussion". Epigenetics in cancer is of course an extremely complex issue stretching far beyond what is mentioned here. 

The issue of comorbidity and its potential contribution to patient variable prognosis was added to the "Discussion" (also in response to reviewer 2). 

References

1. Bell CC, Gilan O. Principles and mechanisms of non-genetic resistance in cancer. Br J Cancer. 2020 Feb;122(4):465-472. doi: 10.1038/s41416-019-0648-6. Epub 2019 Dec 13. PMID: 31831859; PMCID: PMC7028722. 

2. Inde Z, Dixon SJ. The impact of non-genetic heterogeneity on cancer cell death. Crit Rev Biochem Mol Biol. 2018 Feb;53(1):99-114. doi: 10.1080/10409238.2017.1412395. Epub 2017 Dec 18. PMID: 29250983; PMCID: PMC6089072.

3. Bell CC, Fennell KA, Chan YC, Rambow F, Yeung MM, Vassiliadis D, Lara L, Yeh P, Martelotto LG, Rogiers A, Kremer BE, Barbash O, Mohammad HP, Johanson TM, Burr ML, Dhar A, Karpinich N, Tian L, Tyler DS, MacPherson L, Shi J, Pinnawala N, Yew Fong C, Papenfuss AT, Grimmond SM, Dawson SJ, Allan RS, Kruger RG, Vakoc CR, Goode DL, Naik SH, Gilan O, Lam EYN, Marine JC, Prinjha RK, Dawson MA. Targeting enhancer switching overcomes non-genetic drug resistance in acute myeloid leukaemia. Nat Commun. 2019 Jun 20;10(1):2723. doi: 10.1038/s41467-019-10652-9. PMID: 31222014; PMCID: PMC6586637.

4. Irish JM, Hovland R, Krutzik PO, Perez OD, Bruserud Ø, Gjertsen BT, Nolan GP. Single cell profiling of potentiated phospho-protein networks in cancer cells. Cell. 2004 Jul 23;118(2):217-28. doi: 10.1016/j.cell.2004.06.028. PMID: 15260991. 

Reviewer #2: The present article tries to prove that the survival of patients across different time points can be predicted and set a new carcinogenesis parameter

Comment 1: The drawbacks of such an assumption among other are:

1) Stage migration due to more sophisticated techniques during last decades.

2) Different treatment efficacy during last decades.

3) Different treatment toxicity during last decades.

4) Different expected population survival due to the improvements in other specialties (e.g. cardiology) during last decades.

Response: Medicine keeps evolving and better diagnostic and therapeutic tools became available in more resent years. This led to stage migration, to more effective and less toxic therapies, and to better care of comorbidities. As the reviewer suggests, this phenomenon can decrease variability and improve the PI of more recently diagnosed patients. This important point was added to the discussion.

---

## [Decision Letter · Decision Letter 1]

27 May 2024

Predictability: A New Distinguishing Feature of Cancer?

PONE-D-24-05545R1

Dear Dr. Gofrit,

We’re pleased to inform you that your manuscript has been judged scientifically suitable for publication and will be formally accepted for publication once it meets all outstanding technical requirements.

Kind regards,

Sina Azadnajafabad, MD, MPH

Academic Editor

PLOS ONE

Additional Editor Comments (optional):

Reviewers' comments:

Reviewer's Responses to Questions

**Comments to the Author**

1. If the authors have adequately addressed your comments raised in a previous round of review and you feel that this manuscript is now acceptable for publication, you may indicate that here to bypass the “Comments to the Author” section, enter your conflict of interest statement in the “Confidential to Editor” section, and submit your "Accept" recommendation.

Reviewer #1: (No Response)

Reviewer #2: All comments have been addressed

2. Is the manuscript technically sound, and do the data support the conclusions?

Reviewer #1: Yes

Reviewer #2: Yes

3. Has the statistical analysis been performed appropriately and rigorously? 

Reviewer #1: Yes

Reviewer #2: Yes

4. Have the authors made all data underlying the findings in their manuscript fully available?

Reviewer #1: Yes

Reviewer #2: Yes

5. Is the manuscript presented in an intelligible fashion and written in standard English?

Reviewer #1: Yes

Reviewer #2: Yes

6. Review Comments to the Author

Reviewer #1: (No Response)

Reviewer #2: All comments have been adressed adequately. Therefore, the manuscript can be further published in the present form

7. PLOS authors have the option to publish the peer review history of their article (what does this mean?). If published, this will include your full peer review and any attached files.

Reviewer #1: No

Reviewer #2: No

---

## [Editor Report · Acceptance letter]

3 Jun 2024

PONE-D-24-05545R1 

PLOS ONE

Dear Dr. Gofrit, 

I'm pleased to inform you that your manuscript has been deemed suitable for publication in PLOS ONE. Congratulations! Your manuscript is now being handed over to our production team.

Kind regards, 

on behalf of

Dr. Sina Azadnajafabad 

Academic Editor

PLOS ONE